# Soft Non-Diagonality Penalty Enables Latent Space-Level Interpretability of pLM at No Performance Cost

## Abstract

Emergence of large scale protein language models (pLMs) has led to significant performance gains in predictive protein modeling. However, it comes at a high price of interpretability, and efforts to push representation learning towards explainable feature spaces remain scarce. The prevailing use of domain-agnostic and sparse encodings in such models fosters a perception that developing both parameter-efficient and generalizable models in a low-data regime is not feasible. In this work, we explore an alternative approach to develop compact models with interpretable embeddings while maintaining competitive performance. With the Bidirectional Long Short-Term Memory Autoencoder (BiLSTM-AE) model as an example trained on positional property matrices, we introduce a soft weight matrix non-diagonality penalty. Through Jacobian analysis, we show that this penalty aligns embeddings with the initial feature space while leading to a marginal increase in performance on a suite of four common peptide biological activity classification benchmarks. Moreover, it was demonstrated that the use of one-hot encoded sequence clustering-based contrastive loss to produce semantically meaningful latent space allows to further improve benchmarking performance. The use of amino acid physicochemical properties and density functional theory (DFT) derived cofactor interaction energies as input features provides a foundation for intrinsic interpretability, which we demonstrate on fundamental peptide properties. The resulting model is over 33,000 times more compact than the state-of-the-art pLM ProtT5. It demonstrates performance stability across diverse benchmarks without task-specific fine-tuning, showcasing that domain-tailored architectural design can yield highly parameter-efficient models with fast inference and preserved generalization capabilities.

## 1 Introduction

Machine learning (ML) has achieved substantial progress in drug discovery (Jorner et al., 2021; Lee et al., 2020), supported by representations such as SMILES (Weininger, 1988), which can be converted into molecular graphs and used to calculate interpretable chemical descriptors using RDKit (Landrum et al., 2013) or Mordred (Moriwaki et al., 2018). However, these representations become limiting for biopolymers due to their higher structural complexity. Peptides, a class of biopolymers composed of <100 amino acids, exhibit various conformations governed by hydrogen bonding, hydrophobic interactions, metal coordination, and disulfide bridges (Gregorc et al., 2023; Rezai et al., 2006; Victorio & Sawyer, 2023). Their larger contact surface and rich interaction patterns make them highly selective and effective therapeutic agents (Henninot et al., 2018; Fisher et al., 2019; Peterson & Barry, 2018; Torres et al., 2019). At the same time, their biological activity depends on a sequence context that is difficult to capture with small-molecule–oriented descriptors.

ML methods are widely used in peptide research (Basith et al., 2020), and peptide representations typically follow two directions: property-based and sequence-based (Xu et al., 2020). The first relies on physicochemical descriptors and provides interpretability, but does not account for positional context. The second captures sequence-level information, yet lacks explicit chemical grounding and interpretability. In practice, models built on either approach tend to be task-specific. This

motivates the need for representations that combine contextual modeling with chemically meaningful, interpretable structure.

Using a Bidirectional Long Short-Term Memory Autoencoder (BiLSTM-AE) trained on positional property matrices, we introduce a soft penalty on off-diagonal weight-matrix elements together with diagonal initialization. This encourages feature disentanglement and aligns latent dimensions with the original physicochemical feature space. Jacobian-based analysis confirms that the penalty enforces this alignment while slightly improving performance across four peptide biological activity classification benchmarks.

We further show that incorporating one-hot encoded sequences clustering-based contrastive loss produces more semantically structured latent spaces, leading to additional performance gains. Because the model operates directly on amino acid physicochemical properties, including interaction energies derived from density functional theory (DFT), it naturally supports intrinsic interpretability. We illustrate this using regression tasks on fundamental peptide physicochemical properties.

To more fairly compare embeddings across benchmarking datasets, we introduce average Shannon entropy and Levenshtein distance based measures to approximate sequence and dataset complexity, since dataset size alone does not reliably predict model performance. Despite its compactness - being >4,000, >12,000, and >6,000 times more compact than ProtBERT (Elnaggar et al., 2022), Ankh-large (Elnaggar et al., 2023), and ESM-C (600M) (ESM Team, 2024), respectively - our model achieves competitive results. It surpasses all baselines on anti-inflammatory peptide prediction (AIP) , matches Ankh-large on antimicrobial peptides (AMP), performs comparably to ESM-C on anti-oxidative peptides (AOP), and outperforms ProtBERT on AOP and anti-diabetic peptide prediction (ADP) . Based on Matthew's correlation coefficient (MCC), it also shows the second-highest performance stability across all tasks after Ankh-large.

Overall, our model is more than 33,000 times more compact than the state-of-the-art (SOTA) protein language model (pLM) ProtT5-3B (Elnaggar et al., 2022)and achieves stable performance across diverse peptide tasks without task-specific fine-tuning. This illustrates that domain-tailored architectures can yield parameter-efficient models with fast inference and strong generalization. At the same time, scaling such interpretable models remains a challenge and warrants further investigation to close the performance gap with the largest protein language models.

**Key contributions**:

- We introduce a soft penalty on off-diagonal weight-matrix elements combined with diagonal initialization in a BiLSTM-AE, enabling feature disentanglement and alignment of latent dimensions with amino acid physicochemical properties.
- We demonstrate latent-space interpretability through feature importance and correlation analysis in four regression tasks involving fundamental peptide physicochemical properties.
- We show that the resulting model achieves competitive or superior performance to ProtBERT, Ankh-large, and ESM-C on four peptide biological activity classification benchmarks while remaining intrinsically interpretable.
- We find that embeddings from many existing protein language models still perform comparably to simple one-hot baselines on peptide tasks, highlighting ongoing challenges in peptide-specific representation learning with domain-agnostic large models.
- We show that the proposed non-diagonality penalty contributes to a small but consistent improvement in benchmark performance within our architecture and training setup.
- We demonstrate that our final model is 3–4 orders of magnitude more parameter-efficient than ProtBERT, ESM-C, Ankh, and ProtT5-3B while trained on datasets at least two orders of magnitude smaller and without task-specific fine-tuning.

## 2 RELATED WORKS

Current SOTA pLMs are largely based on the Transformer architecture (Vaswani et al., 2017), adapted to biological sequences, and pre-trained on massive protein corpora. Representative models such as ProtBERT and ProtT5 extend the BERT (Devlin et al., 2019) and T5 (Raffel et al., 2020) architectures to enable effective transfer learning across diverse protein tasks. The ESM family further scales

this paradigm, with models like ESM-Cambrian leveraging hundreds of millions of parameters to capture evolutionary and structural regularities at large scale. The Ankh work systematically explores architectural design choices: span masking, activation functions, positional encodings, to optimize Transformer-based pLMs for protein modeling. Despite architectural differences, these models share an important limitation: their interpretability relies primarily on analyzing self-attention patterns. Although attention maps can highlight residue–residue interactions or structural contacts, they are often noisy, vary widely between layers and heads, and do not reliably correspond to causal importance (Jain & Wallace, 2019).

Motivated by this limitation, a variety of post-hoc interpretability methods have been developed. Attribution techniques, such as attention analysis, probing, and gradient-based methods, aim to uncover sequence elements that influence model outputs but do not alter or structure the latent space itself (Hunklinger & Ferruz, 2025). Sparse autoencoders (SAEs) offer a recent alternative by training sparse decoders on pLM activations to extract biologically meaningful latent factors (Gujral et al., 2025; Simon & Zou, 2025). These methods reveal that pLM representations encode useful but highly entangled structure. As a result, both attribution-based approaches and SAEs provide useful insight but remain fundamentally post-hoc and limited in their ability to shape embedding space.

In contrast, intrinsically interpretable peptide representation learning remains relatively underexplored. Existing approaches often sacrifice contextual modeling capacity or rely heavily on manual feature engineering. Our work addresses this gap by introducing an inductive bias directly into a sequence model: a soft non-diagonality penalty that encourages latent dimensions to align with predefined physicochemical feature axes. This design embeds interpretability into the model itself, producing disentangled chemically grounded representations while preserving the contextual expressiveness of recurrent architectures.

## 3 METHODS

### 3.1 CONTEXT AND MOTIVATION

The proposed method constructs property matrices for peptides from physico-chemical descriptors of 20 canonical residues and 2 prevalent modifications. It uses 43 molecular features and 3 DFT-derived. Convolutional autoencoders (CAE) serve as a baseline, while recurrent and transformer variants are introduced to capture long-range dependencies. The embeddings are further refined using Information Noise-Contrastive Estimation (InfoNCE) based contrastive loss. Positive and negative pairs are generated via *MiniBatchKMeans* clustering of one-hot encoded sequences, ensuring that the interpretable property space corresponds to empirically effective sequence encodings. A soft non-diagonality penalty guides embedding features toward the original physicochemical space, with feature relationships quantified using the Jacobian matrix (see **Section 3.7**). An overview of the full processing pipeline is shown in **Figure 1**.

### 3.2 DATA COLLECTION

For model training, we retrieved unlabeled unique peptide sequences containing no extra monomers from the NCBI database (Sayers et al., 2023), yielding 6,749,334 sequences. A sampling procedure is implemented to reduce data volume and computational costs while preserving diversity: peptides are grouped by the most frequently occurring amino acids. This allowed us to maintain balanced composition over sequence length distribution. Consequently, small (**S**) and big (**B**) datasets are constructed through clustering peptide sequences based on amino acid frequency vectors, with the initial number of clusters equal to the unique amino acids. Then, the unique clusters are identified on the basis of the most represented residues for stratified sampling; the minimum number of samples is determined to prevent under-representation of smaller clusters. A small fraction (1%) of sequences forms dataset **S**; a medium fraction (10%) is sampled for dataset **B**. Finally, the subsets are combined to create the final datasets. Cluster-based stratified sampling facilitated the formation of manageable datasets, where the final model is trained using the dataset **S** comprising 155,920 sequences. Final data scaling experiment is performed on dataset **B** comprising 467,792 sequences.

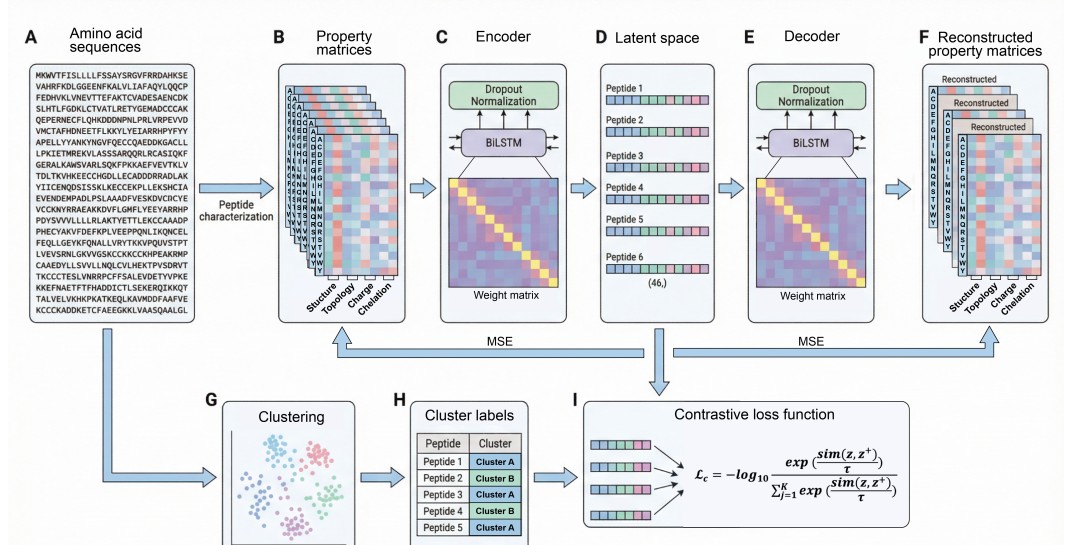

Figure 1: Overview of the proposed pipeline.

### 3.3 MOLECULAR DESCRIPTORS

A total of 43 molecular RDKit descriptors are used to characterize the properties of amino acids (see **Appendix A**), which were normalized to a range of [-1, 1]. Each peptide sequence is represented as a feature array based on these descriptors and DFT features (see **Section 3.4** for details), with padding applied to align all sequences to a uniform length of 96 amino acids.

### 3.4 DFT-DERIVED DESCRIPTORS

This group of descriptors includes three parameters derived from the interactions energies of amino acids with divalent metal cations: calcium ($Ca^{2+}$), magnesium ($Mg^{2+}$), and barium ($Ba^{2+}$), obtained through DFT calculations. Detailed methodology is described in Hu et al. (2022), with data available in the NOMAD database (Draxl & Scheffler, 2019). The free energy values are extracted using the NOMAD API and categorized into four groups: complexes with $Mg^{2+}$, $Ca^{2+}$, and $Ba^{2+}$ cations and isolated amino acid conformers. For each amino acid, average free energy values across all conformers is calculated, and interaction descriptors are derived by normalizing the differences between complex and isolated free energy values.

### 3.5 AE-BASED ARCHITECTURES

**Convolutional and Variational Autoencoders (CAE/VAEs).** CAE-based models with per-feature convolutions were treated as a baseline to study the model performance when features were not mixed at all due to architectural restrictions. We then implemented a probabilistic framework by replacing the deterministic latent space of CAE with a variational latent space. This approach encourages smoother, more continuous latent representations, potentially mitigating sparsity. Experiments were conducted with multiple VAE variants, including $\beta$-VAE and InfoVAE. Hyperparameters, including the KL divergence weight ($\beta$), were chosen from a predetermined grid, this ensured an unbiased evaluation of embedding quality.

**BiLSTM.** Given the sequential nature of the data, BiLSTM-based architectures were used to capture long-range dependencies. Unlike unidirectional recurrent models, BiLSTM leverages both forward and backward contexts, enabling a more comprehensive understanding of sequence information. We hypothesized that this bidirectional approach would enhance the quality of the generated embeddings, making them more effective for downstream tasks.

**Transformer-Based Models.** Motivated by the success of transformer architectures in natural language processing, we tested a transformer-based AE. Transformers leverage self-attention mechanisms to capture global dependencies, typical in peptide sequences.

Each of the architectures described in this section was trained using the same dataset **S** and evaluated on identical downstream tasks to ensure a fair comparison with the benchmark encoding strategies (see **Appendix B**).

### 3.6 CONTRASTIVE LEARNING IMPLEMENTATION

Contrastive learning was implemented to enhance embedding quality by drawing positive pairs closer in latent space and separating negative pairs.

**Positive and Negative Pair Construction.** Given the unique properties of peptide sequences, we designed a domain-specific strategy for constructing positive and negative pairs:

- Positive Pairs: Positive pairs were obtained by *MiniBatchKMeans* clustering of one-hot encoded peptides; k was selected *via* the Elbow criterion, and peptides sharing a cluster were deemed positives.
- Negative Pairs: Peptides from different clusters were treated as negatives.

**Contrastive Loss Function.** Utilized contrastive loss is based on the InfoNCE objective:

$$\mathcal{L}_{\text{contrastive}} = -\log \frac{\exp(\frac{\text{sim}(\mathbf{z},\mathbf{z}^+)}{\tau})}{\sum_{j=1}^{K} \exp(\frac{\text{sim}(\mathbf{z},\mathbf{z}_j)}{\tau})}, \tag{1}$$

where $\mathbf{z} = E(\mathbf{x})$, $E(\cdot)$ is the encoder, $\text{sim}(\cdot, \cdot)$ denotes a similarity measure (e.g., cosine similarity), $K$ is the number of negative samples in a batch, and $\tau > 0$ is a temperature hyperparameter.

### 3.7 SOFT NON-DIAGONALITY PENALTY

For the BiLSTM architecture described above, we introduce an additional regularization mechanism to reduce feature entanglement during training.

To achieve this, two complementary techniques are employed. First, we apply a soft penalty specifically to the off-diagonal elements of the BiLSTM weight matrices. For a weight matrix $M \in \mathbb{R}^{n \times n}$, the penalty term is defined as the mean of squared off-diagonal elements:

$$\mathcal{L}_M = \frac{\|M \odot (\mathbf{1} - I)\|_F^2}{n \cdot n} \tag{2}$$

where $\odot$ denotes the Hadamard product, $I$ is the identity matrix, and $\mathbf{1}$ is a matrix of ones. The total penalty is the sum of all weight matrices, scaled by a coefficient $\lambda$:

$$\mathcal{L}_{\text{off-diag}} = \lambda \sum_{M \in \mathcal{W}} \mathcal{L}_M \tag{3}$$

Here, $\mathcal{W}$ comprises all *input-to-hidden* and *hidden-to-hidden* weight matrices of the BiLSTM layers. Second, we use forced diagonal weight initialization as an inductive bias. The combined effect encourages the model to maintain nearly diagonal weight transformations throughout training.

We additionally evaluated hard diagonalization - strict constraints enforcing zero off-diagonal weights. Although theoretically appealing, this formulation restricted the optimization to an extremely narrow feasible set, preventing stable convergence of the reconstruction and contrastive objectives. Because these difficulties occurred during optimization itself, we did not pursue the hard constraint further. The soft formulation preserves differentiability and allows the optimization to proceed while still encouraging near-diagonal structure.

To quantitatively evaluate the degree of feature disentanglement, we propose a diagonality metric based on the encoder's Jacobian. We compute the mean absolute Jacobian $\bar{J} \in \mathbb{R}^{D_{in} \times D_{out}}$ over the dataset. The diagonality metric is then defined as:

$$\mathcal{D} = \frac{\sum_i \bar{J}_{ii}}{\sum_{i,j} \bar{J}_{ij}} \tag{4}$$

A value of $\mathcal{D}$ close to 1 indicates a strong feature-wise separation.

### 3.8 COMPUTATIONAL RESOURCES

All calculations were performed on a server with a general configuration consisting of 6 A6000 GPUs, 256 cores, AMD EPYC 7763 64-Core Processor, 512 GB RAM. The training procedure was performed using 2 GPUs and 50 GB RAM.

## 4 EXPERIMENTS

### 4.1 BENCHMARK DATASETS

We performed benchmarking of all the models mentioned in the paper on four public peptide biological activity classification datasets: antimicrobial (AMP) (Cao et al., 2023), anti-inflammatory (AIP) (Raza et al., 2023), antidiabetic (ADP) (Chen et al., 2022), and antioxidant (AOP) (Qin et al., 2023). Dataset statistics covering size, sequence complexity, and data imbalance are summarized in **Table 1**. Sequence complexity is described here by several parameters including length statistics, as well as average Shannon entropy characterizing the extent of non-equiprobability of amino acids per sequence and average Levenshtein distance showing inter-sequence dissimilarity. Based on these statistics, ADP appears to be the most challenging dataset based on dataset size and high values of average Shannon entropy, Levenshtein distance, as well as length spread.

Table 1: Main statistics of benchmark datasets. Average Levenshtein distance characterizes the diversity of sequences by measuring the pairwise differences. Average Shannon entropy characterizes the diversity of amino acids within a single sequence by quantifying the average level of uncertainty associated with all types of amino acids.

| Dataset | Num. of peptides | Avg. Levenshtein distance | Avg. Shannon entropy | Avg. length | Min. length | Max. length | Pos/Neg sample ratio |
|---------|------------------|---------------------------|----------------------|-------------|-------------|-------------|----------------------|
| ADP | 472 | $19.20 \pm 4.26$ | $3.22 \pm 0.16$ | $19.60 \pm 5.09$ | 11 | 41 | 1.00 |
| AIP | 3790 | $15.48 \pm 2.36$ | $3.15 \pm 0.11$ | $16.39 \pm 2.62$ | 11 | 30 | 0.76 |
| AMP | 8268 | $18.38 \pm 4.21$ | $3.03 \pm 0.26$ | $18.53 \pm 5.34$ | 11 | 30 | 1.00 |
| AOP | 2120 | $7.12 \pm 3.58$ | $2.03 \pm 0.48$ | $5.92 \pm 3.66$ | 2 | 20 | 1.00 |

### 4.2 EXPERIMENTAL SETUP

This section presents the experimental framework for evaluating the quality and generalizability of the embeddings generated by our model. The assessment was conducted on four peptide classification datasets encompasing distinct biological activities (detailed in **Section 4.1**). To isolate the evaluation on the embedding quality itself, rather than the complexity of a deep learning classifier, we employed a classical gradient boosting model as a simple yet effective downstream predictor (Shwartz-Ziv & Armon, 2022). The primary metric for comparison was the MCC, chosen for its informativeness and robustness to class imbalance (Chicco & Jurman, 2020) — a property particularly relevant for datasets like AIP, as evidenced by the statistics in **Table 1**. For reliable model assessment, we employed a rigorous 5-fold cross-validation protocol, reporting both the mean metric values and their standard errors (the latter indicated in parentheses).

Our comparative analysis includes three conventional peptide encoding methods: one-hot encoding, BLOSUM62, and 3-mer counts, and several SOTA pLMs: ProtBERT, Ankh-large, ESM-C (600M), and ProtT5-3B. The following subsections detail the development and optimization experiments for our model, with results provided in **Appendixes C** and **F** (Tables 6 through 14). The final comparative benchmarking results against all baselines are consolidated in **Section 4.5** (**Table 2**).

### 4.3 AE-BASED ARCHITECTURES BENCHMARKING

We begin by establishing a reference using a CAE with per-feature convolutions, which serves as a baseline where no feature mixing is possible due to the architectural structure. Results for all encodings with the CAE model are shown in **Appendix C**.

To perform a systematic comparison, we evaluated a pre-specified set of 80 models across four AE families: CAEs, VAEs (incl. Info-VAE, $\beta$-VAE), BiLSTM-AEs, and transformer-AEs—each trained under a fixed grid of hyperparameters (**Appendix B**). The embeddings were evaluated with the same downstream predictor as the baseline and benchmark encodings (see **Section 4.2**). Subsequent methodological developments were implemented using the BiLSTM-AE architecture due to its suitability for sequential inputs and stable training dynamics (see **Table 7**).

### 4.4 CLUSTERING-BASED CONTRASTIVE LEARNING

We next incorporated a contrastive learning component (**Section 3.6**) to explicitly structure the latent space by drawing similar peptides closer while pushing dissimilar ones apart. The comparison results for BiLSTM-AE (cBiLSTM-AE) with the benchmark encodings appear in **Table 2**.

### 4.5 SOFT NON-DIAGONALITY PENALTY

To enforce feature disentanglement within the model, we applied a soft penalty to the off-diagonal elements of the weight matrices (as detailed in **Section 3.7**). Our evaluation includes two main parts: (1) a comprehensive benchmarking of cBiLSTM-AE model with soft non-diagonality penalty (dcBiLSTM-AE) against all baselines (see **Section 4.2** for details), and (2) a scalability analysis where the model was trained on a large-scale dataset **B**. The results of both the comparative benchmarking and the scalability experiment are consolidated in **Table 2**.

Table 2: dcBiLSTM-AE model benchmarking. No preliminary fine-tuning on tasks was performed for any of the models.

| Encoding type | Model size | Dataset for training | MCC (5-fold cross validation) | | | | Avg. | Min | Max |
|---|---|---|---|---|---|---|---|---|---|
| | | | Anti diabetic | Anti inflammatory | Anti microbial | Anti oxidant | | | |
| One-hot | N/A | N/A | 0.197 (0.015) | 0.216 (0.015) | 0.560 (0.004) | 0.764 (0.013) | 0.434 | 0.197 | 0.764 |
| Blosum | N/A | N/A | 0.026 (0.020) | 0.290 (0.012) | 0.337 (0.015) | 0.189 (0.012) | 0.211 | 0.026 | 0.337 |
| Threemers | N/A | N/A | 0.131 (0.060) | **0.357** **(0.009)** | 0.519 (0.003) | 0.539 (0.015) | 0.387 | 0.131 | 0.539 |
| ProtBert | 420M | 217M | 0.334 (0.031) | 0.138 (0.021) | 0.658 (0.007) | 0.580 (0.011) | 0.428 | 0.138 | 0.658 |
| ESM C | 600M | - | 0.433 0.017 | 0.193 (0.012) | **0.679** **(0.005)** | 0.761 (0.019) | 0.517 | 0.193 | 0.761 |
| Ankh | 1.15B | 59M | **0.574)** **(0.032)** | 0.335 (0.011) | 0.614 (0.007) | **0.863** **(0.015)** | **0.596** | 0.335 | **0.863** |
| BiLSTM-AE | 1.3M | **0.15M** | -0.004 (0.053) | 0.342 (0.009) | 0.415 (0.005) | 0.496 (0.031) | 0.312 | -0.004 | 0.496 |
| cBiLSTM-AE | 1.3M | **0.15M** | 0.277 (0.065) | 0.316 (0.009) | 0.569 (0.002) | 0.692 (0.012) | 0.464 | 0.277 | 0.692 |
| dcBiLSTM-AE | **90K** | **0.15M** | 0.371 (0.039) | 0.355 (0.004) | 0.611 (0.006) | 0.775 (0.018) | 0.528 | **0.355** | 0.775 |
| dcBiLSTM-AE | **90K** | 0.47M | 0.327 (0.042) | 0.337 (0.009) | 0.599 (0.009) | 0.750 (0.016) | 0.503 | 0.327 | 0.750 |
| ProtT5 | 3B | 45M | **0.659** **(0.029)** | **0.402** **(0.013)** | **0.686** **(0.008)** | **0.893** **(0.007)** | **0.660** | **0.402** | **0.893** |

### 4.6 DCBiLSTM-AE EMBEDDINGS CORRELATION ANALYSIS

To assess representational fidelity, we computed four physicochemical properties on the dataset **S** test split (see **Section 3.2**). Targets were chosen to be analytically computable for interpretation clarity,

yet being functionally relevant: instability index (ISI), theoretical net charge (TNC), isoelectric point (IEP), and molecular weight (MW). Correlating these clear whole-peptide properties with interpretable feature space reveals the extent to which the embeddings capture monomer-level physicochemical structure. (**Table 3**). The same protocol was applied to earlier encodings (**Table 14**).

# 5 DISCUSSION

## 5.1 MODEL SCREENING AND OPTIMIZATION

We initially benchmarked common peptide encoding strategies along with the baseline CAE model on four datasets (**Table 1**) to characterize task complexity and establish baseline performance. AMP and AOP datasets were generally easier for the models to classify (MCC = 0.499 and 0.503, respectively; **Table 6**), while ADP and AIP proved more challenging (MCC = 0.165 and 0.267). This difficulty correlates with high Shannon entropies and large inter-sequence Levenshtein distances, combined with relatively small dataset sizes. The baseline CAE achieved moderate performance (**Table 6**): fourth out of five in AMP and AOP, third in ADP, and second in AIP. Notably, CAE performed relatively better on the more challenging datasets, achieving comparable rankings to ProtBERT.

To improve embedding quality, we evaluated architectures designed for sequential data, including BiLSTM and transformer-based autoencoders. Additionally, we tested the InfoCVAE model (Zhao et al., 2019), which addresses potential latent space sparsity and mitigates the tendency of VAEs to ignore latent variables when using flexible decoders. The results of the AE-based models screening stage are summarized in **Table 7**. CAE and BiLSTM-AE showed comparable performance, while the transformer-based AE was less stable during training. The BiLSTM-AE was selected for subsequent experiments due to its architectural suitability for sequential inputs and stable training dynamics.

We incorporated contrastive learning into the training process (details in **Section 3.6**) to further structure the latent space, using one-hot encoded sequences as an initial reference. As demonstrated in **Table 2**, contrastive learning increased the model's predictive metrics compared with both the one-hot baseline and the original BiLSTM-AE.

To mitigate feature entanglement, we implemented a feature disentanglement procedure (**Section 3.7**), combining forced diagonal initialization with a soft penalty. Although this structural constraint could potentially reduce predictive capacity, the results in **Table 2** indicate that performance was slightly improved. This suggests that the imposed structural bias acts as a useful regularizer, effectively guiding the learning process. This procedure also enabled interpretability of the latent space, quantified by a diagonality metric of 49.4% for the dcBiLSTM-AE model (**Section 3.7**).

These results demonstrate that our model consistently delivers stable and robust performance across the evaluated cases including ADPs and AIPs being the most problematic for all encoding strategies under the study, while showing latent space-level interpretability. For instance, although dominance in raw performance is not the main aim of this research, in two out of four tasks our method achieved higher MCC than the ProtBERT model. This result is particularly remarkable given that our model has no preliminary fine-tuning step (thereby, excluding a trivial explanation that the model performs well due to overfitting) and operates with >4,000 times fewer trainable parameters. This drastic reduction in model complexity underscores the efficiency, while still achieving competitive and reliable performance across diverse tasks. Detailed discussion of the model's limitations and potential improvements is provided in **Appendix D**.

## 5.2 EMBEDDING SPACE INTERPRETABILITY STUDIES

To evaluate the interpretability of the obtained peptide embeddings, we conducted a regression analysis aimed at predicting a set of basic peptide properties (see **Section 4.5**). Additionally, we assessed the relationships between individual embedding elements and several biologically significant physicochemical properties of peptides using Pearson's correlation tests. The embeddings generated by our model were used as descriptors for an extreme gradient boosting regression model, which predicted the target properties based on these embeddings. The regression analysis demonstrated that the embeddings effectively captured relevant information about the peptide properties, where the results for the final dcBiLSTM-AE model were comparable with the other approaches. Detailed results of this analysis are provided in **Table 9** (**Appendix C**). To further explore the interpretability

of the embeddings, we examined the correlations between individual latent space features as well as the physicochemical properties. For each property, we identified the top 10 embedding elements with the highest absolute Pearson correlation coefficients (PCC). Among these, we highlighted the features with the most statistically significant correlations based on their p-values.

Overall, the analysis revealed that certain features consistently demonstrated strong correlations with specific physicochemical properties (**Table 3**; for a complete list see **Appendix F**), which can be interpreted as indirect yet a strong sign that the latent space features provide a substantial extent of interpretability. For instance, results for ISI show the strongest correlation of number of aromatic rings, saturated heterocycles and spiro atoms with the target value. Each of these parameters characterize connectivity within the molecules, therefore being related to branching, which directly influences peptide conformational flexibility and, therefore, the stability. Also, in case of TNC and IEP directly related to peptide charge, model shows that partition coefficient logP and $Ca^{2+}$ interaction energy are highly relevant, where the former is a parameter describing protein solubility and the latter is strongly related to the presence and alignment of negatively charged amino acids in the protein. It is important to note, however, that not all the features have a clear and well-studied relationship with the properties predicted in these tasks, while for some of the tasks (e.g., MW), most of the parameters are directly related to the target value, and the model was unable to differentiate between them. Overall, these findings suggest that our model is able to capture meaningful features related to peptide properties, providing a foundation for interpreting its outputs in a biologically informed manner.

Table 3: Correlation between embedding dimensions and peptide properties.

| Peptide property | Encoded input feature | PCC (r) |
|---|---|---|
| ISI | Num Arom Rings | -0.253 |
| | Num Saturated Heterocycles | 0.202 |
| | Num Spiro Atoms | -0.193 |
| TNC | CrippenClogP | -0.738 |
| | $Ca^{2+}$ interaction energy | 0.542 |
| | Num Atom Stereo Centers | -0.537 |
| IEP | CrippenClogP | -0.620 |
| | $Ca^{2+}$ interaction energy | 0.494 |
| | Num Atom Stereo Centers | -0.392 |
| MW | chi1v | -0.994 |
| | chi2v | 0.992 |
| | kappa3 | 0.989 |

## 6 CONCLUSION

In this work, we propose a novel approach towards domain-tailored development of 3-4 orders more parameter-efficient pLMs trained on much smaller yet representative peptide datasets combining cross-task stability requiring no per-task fine-tuning and feature space-level interpretability for peptide representation learning. Although model scaling requires further research to compete with or surpass current SOTA ProtT5 model with input-level interpretability in raw performance, these findings hold great potential by introducing explainable latent spaces crucial for domain scientists building their own predictive models on specific downstream tasks, which is not provided by any of the pLM models existing to date, at no performance cost.

### REPRODUCIBILITY STATEMENT

The code and datasets used in this study are available to ensure reproducibility of the presented results. They can be accessed at the following repository: `https://anonymous.4open.science/r/SeQuant-perfomance-1822/`.

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

# A    APPENDIX A

Table 4: Full list of chemical descriptors used for monomer-wise peptide sequence description.

| Property | Descriptors |
|---|---|
| Mass | exactmw, amw |
| Lipophilicity and Solubility | CrippenClogP, CrippenMR |
| Hydrogen bonds | lipinskiHBA, lipinskiHBD, NumRotatableBonds, NumHBD, NumHBA |
| Structural descriptors | NumHeavyAtoms, NumAtoms, NumHeteroatoms, NumAmideBonds, FractionCSP3 |
| Ring Structures | NumRings, NumAromaticRings, NumAliphaticRings, NumSaturatedRings, NumHeterocycles, NumAromatic Heterocycles, NumSaturatedHeterocycles, NumAliphatic Heterocycles, NumSpiroAtoms, NumBridgeheadAtoms |
| Topological indices | chi0v, chi1v, chi2v, chi3v, chi4v, chi0n, chi1n, chi2n, chi3n, chi4n, hallKierAlpha, kappa1, kappa2, kappa3 |
| Polar surface | tpsa |
| Surface area | labuteASA |
| Energy | $Ca^{2+}$ Interaction energy, $Mg^{2+}$ Interaction energy, $Ba^{2+}$ Interaction energy |
| Other descriptors | Phi |

# B APPENDIX B

Table 5: Hyperparameter configurations for experimental model variants. 1-30 CAE; 31-49 VAE; 50 Transformer; 51 GRU; 52 LSTM; 53 BiLSTM; 54 BiLSTM + attention; 55-80 cBiLSTM. Horizontal lines stands for different experimental stages, details are provided in Comment column.

| Architecture | Trainable parameters | Hyperparameters | Comment |
|---|---|---|---|
| 1 | 511943 | batch_size = 64
filters = 46
pool_size = 3-2-2-2-2-2
depth (number of layer blocks) = 6
loss_function = MSE | Number of filters is constant |
| 2 | 511943 | batch_size = 64
filters = 46
pool_size = 2-2-2-2-2-3
depth (number of layer blocks) = 6
loss_function = MSE | Number of filters is constant |
| 3 | 61536 | batch_size = 64
filters = 46
pool_size = 4-3-2-2-2
depth (number of layer blocks) = 5
loss_function = MSE | Number of filters is constant |
| 4 | 61536 | batch_size = 64
filters = 46
pool_size = 2-2-2-3-4
depth (number of layer blocks) = 5
loss_function = MSE | Number of filters is constant |
| 5 | 42500 | batch_size = 64
filters = 46
pool_size = 8-3-2-2
depth (number of layer blocks) = 4
loss_function = MSE | Number of filters is constant |
| 6 | 42500 | batch_size = 64
filters = 46
pool_size = 4
depth (number of layer blocks) = 2-2-3-8
loss_function = MSE | Number of filters is constant |
| 7 | 735000 | batch_size = 64
filters = 368
pool_size = 3-2-2-2-2-2
depth (number of layer blocks) = 6
loss_function = MSE | Filters decrease: 368-184-92-46-23-1 |
| 8 | 1809052 | batch_size = 64
filters = 368
pool_size = 3-2-2-2-2-2
depth (number of layer blocks) = 6
loss_function = MSE | Filters decrease: 368-368-184-92-46-1. Optimal architecture in the first stage |

| | | | |
|---|---|---|---|
| 9 | 461482 | batch_size = 64
filters = 184
pool_size = 3-2-2-2-2-2
depth (number of layer blocks) = 6
loss_function = MSE | Filters decrease:
184-184-92-46-23-1 |
| 10 | 257000 | batch_size = 64
filters = 184
pool_size = 3-2-2-2-2-2
depth (number of layer blocks) = 6
loss_function = MSE | Filters decrease:
184-92-92-46-23-1 |
| 11 | 148000 | batch_size = 64
filters = 92
pool_size = 3-2-2-2-2-2
depth (number of layer blocks) = 6
loss_function = MSE | Filters decrease:
92-92-46-46-23-1 |
| 12 | 205000 | batch_size = 64
filters = 184
pool_size = 3-2-2-2-2-2
depth (number of layer blocks) = 6
loss_function = MSE | Filters decrease:
184-92-46-23-11-1 |
| 13 | 2901000 | batch_size = 64
filters = 736
pool_size = 3-2-2-2-2-2
depth (number of layer blocks) = 6
loss_function = MSE | Filters decrease:
736-368-184-92-46-1 |
| 14 | 133 463 996 | batch_size = 64
filters = 368
pool_size = 3-2-2-2-2-2
depth (number of layer blocks) = 6
loss_function = MSE | Add attention layers to encoder |
| 15 | 265118940 | batch_size = 64
filters = 368
pool_size = 3-2-2-2-2-2
depth (number of layer blocks) = 6
loss_function = MSE | Add attention layers to encoder and decoder |
| 16 | 1809052 | batch_size = 48
filters = 368
pool_size = 3-2-2-2-2-2
depth (number of layer blocks) = 6
loss_function = MSE | Decrease batch size |
| 17 | 1809052 | batch_size = 32
filters = 368
pool_size = 3-2-2-2-2-2
depth (number of layer blocks) = 6
loss_function = MSE | Decrease batch size |
| 18 | 1809052 | batch_size = 24
filters = 368
pool_size = 3-2-2-2-2-2
depth (number of layer blocks) = 6
loss_function = MSE | Decrease batch size |

| 19 | 1809052 | batch_size = 16
filters = 368
pool_size = 3-2-2-2-2-2
depth (number of layer blocks) = 6
loss_function = MSE | Decrease batch size |
|---|---|---|---|
| 20 | 4337672 | batch_size = 32
filters = 368
pool_size = 3-2-2-2-2-2
depth (number of layer blocks) = 6
loss_function = MSE | Additional Conv2D layers in each block |
| 21 | 8982890 | batch_size = 32
filters = 460
pool_size = 3-2-2-2-2-2
depth (number of layer blocks) = 6
loss_function = MSE | Increase number of filters |
| 22 | 94469672 | batch_size = 32
filters = 368
pool_size = 3-2-2-2-2-2
depth (number of layer blocks) = 6
loss_function = MSE | Add 7 Dense layers to latent space |
| 23 | 62295982 | batch_size = 32
filters = 368
pool_size = 3-2-2-2-2-2
depth (number of layer blocks) = 6
loss_function = MSE | Add 2 Dense layers to latent space |
| 24 | 30 122 292 | batch_size = 32
filters = 368
pool_size = 3-2-2-2-2-2
depth (number of layer blocks) = 6
loss_function = MSE | Add 1 Dense layer to latent space |
| 25 | 4685330 | batch_size = 32
filters = 46
pool_size = 96
depth (number of layer blocks) = 1
loss_function = MSE | |
| 26 | 4702534 | batch_size = 32
filters = 46
pool_size = 48-2
depth (number of layer blocks) = 2
loss_function = MSE | |
| 27 | 4719738 | batch_size = 32
filters = 46
pool_size = 24-2-2
depth (number of layer blocks) = 3
loss_function = MSE | |
| 28 | 4736942 | batch_size = 32
filters = 46
pool_size = 12-2-2-2
depth (number of layer blocks) = 4
loss_function = MSE | |

| | | | |
|---|---|---|---|
| 29 | 4754146 | batch_size = 32
filters = 46
pool_size = 6-2-2-2-2
depth (number of layer blocks) = 5
loss_function = MSE | |
| 30 | 178490 | batch_size = 32
filters = height * ((depth - i) / 2)
pool_size = 3-2-2-2-2-2
depth (number of layer blocks) = 6
loss_function = MSE | Dependence of the number of filters on the depth - Baseline CAE |
| 31 | 176328 | batch_size = 32
filters = height * ((depth - i) / 2)
pool_size = 3-2-2-2-2-2
depth (number of layer blocks) = 6
loss_function = MSE + KL divergence | Add KL divergence |
| 32 | 176328 | batch_size = 32
filters = height * ((depth - i) / 2)
pool_size = 3-2-2-2-2-2
depth (number of layer blocks) = 6
loss_function = MSE + Wasserstein distance | Wasserstein distance |
| 33 | 176328 | batch_size = 32
filters = height * ((depth - i) / 2)
pool_size = 3-2-2-2-2-2
depth (number of layer blocks) = 6
loss_function = MSE + Shannon divergence | Shannon divergence |
| 34 | 176328 | batch_size = 32
filters = height * ((depth - i) / 2)
pool_size = 3-2-2-2-2-2
depth (number of layer blocks) = 6
loss_function = MSE + KL divergence | |
| 35 | 176328 | batch_size = 32
filters = height * ((depth - i) / 2)
pool_size = 3-2-2-2-2-2
depth (number of layer blocks) = 6
loss_function = MSE + KL divergence | Add Gaussian noise |
| 36 | 176328 | batch_size = 32
filters = height * ((depth - i) / 2)
pool_size = 3-2-2-2-2-2
depth (number of layer blocks) = 6
loss_function = MSE + 100 * KL divergence | |
| 37 | 176328 | batch_size = 32
filters = height * ((depth - i) / 2)
pool_size = 3-2-2-2-2-2
depth (number of layer blocks) = 6
loss_function = MSE + 1000 * KL divergence | |

| | | | |
|---|---|---|---|
| 38 | 483010 | batch_size = 32
filters = height * ((depth - i) / 2)
pool_size = 3-2-2-2-2-2-1-1
depth (number of layer blocks) = 8
loss_function = MSE + KL divergence | |
| 39 | 1027236 | batch_size = 32
filters = height * ((depth - i) / 2)
pool_size = 3-2-2-2-2-2-1-1-1-1
depth (number of layer blocks) = 10
loss_function = MSE + KL divergence | |
| 40 | 1876718 | batch_size = 32
filters = height * ((depth - i) / 2)
pool_size = 3-2-2-2-2-2-1-1-1-1-1-1
depth (number of layer blocks) = 12
loss_function = MSE + KL divergence | |
| 41 | 2028048 | batch_size = 32
filters = height * ((depth - i) / 2)
pool_size = 3-2-2-2-2-2-1-1
depth (number of layer blocks) = 8
loss_function = MSE + KL divergence | |
| 42 | 3086554 | batch_size = 32
filters = height * ((depth - i) / 2)
pool_size = 3-2-2-2-2-2-1-1-1-1
depth (number of layer blocks) = 10
loss_function = MSE + KL divergence | |
| 43 | 4 165 944 | batch_size = 32
filters = filters = height * ((depth - i) / 2)
pool_size = 3-2-2-2-2-2-1-1-1-1-1-1
depth (number of layer blocks) = 12
loss_function = MSE + KL divergence | |
| 44 | 176328 | batch_size = 32
filters = height * ((depth - i) / 2)
pool_size = 3-2-2-2-2-2
depth (number of layer blocks) = 6
loss_function = MSE + KL divergence | Change layers order |
| 45 | 9681446 | batch_size = 32
filters = height * ((depth - i) / 2)
pool_size = 3-2-2-2-2-2
depth (number of layer blocks) = 20
loss_function = MSE + KL divergence | |
| 46 | 176328 | batch_size = 32
filters = height * ((depth - i) / 2)
pool_size = 3-2-2-2-2-2
depth (number of layer blocks) = 6
loss_function = MSE + KL divergence | Remove tf.Data.Dataset from data preprocessing |
| 47 | 48882 | batch_size = 32
filters = height * ((depth - i) / 2)
pool_size = 3-2-2-2-2-2
depth (number of layer blocks) = 6
loss_function = MSE + KL divergence | convolution kernel = 2 |

| 48 | 516184 | batch_size = 32
filters = height * ((depth - i) / 2)
pool_size = 3-2-2-2-2-2
depth (number of layer blocks) = 6
loss_function = MSE + KL divergence | convolution kernel = 6 |
|----|--------|---|---|
| 49 | 176328 | batch_size = 32
filters = height * ((depth - i) / 2)
pool_size = 3-2-2-2-2-2
depth (number of layer blocks) = 6
loss_function = MSE + KL divergence + MMD | InfoVAE loss function |
| 50 | 388022 | batch_size = 32
depth (number of layer blocks) = 3
feed forward dim = 128
embeddings dim = 46
loss_function = MSE | Transformer-based model |
| 51 | 515676 | batch_size = 32
depth (number of layer blocks) = 3
loss_function = MSE
gru_units = 256 | GRU-based model |
| 52 | 295452 | batch_size = 32
depth (number of layer blocks) = 3
loss_function = MSE
lstm_units = 160 | LSTM-based model |
| 53 | 256220 | batch_size = 32
depth (number of layer blocks) = 3
loss_function = MSE
bilstm_units = 96 | BiLSTM-based model |
| 54 | 696038 | batch_size = 32
depth (number of layer blocks) = 3
feed forward dim = 128
embeddings dim = 46
loss_function = MSE | BiLSTM-based model with attention layers |
| 55 | 1293788 | batch_size = 32
depth (number of layer blocks) = 3
loss_function = MSE + InfoNCE
bilstm_units = 96 | Add contrastive loss function Clustering on one-hot with PCA data (2D) |
| 56 | 1293788 | batch_size = 32
depth (number of layer blocks) = 3
loss_function = MSE + 0.1 * InfoNCE
bilstm_units = 96 | Add decreasing coefficient for contrastive loss Clustering on one-hot with PCA data (2D) |
| 57 | 1293788 | batch_size = 32
depth (number of layer blocks) = 3
loss_function = MSE + InfoNCE
bilstm_units = 96 | Clustering on one-hot data (without dimensionality reduction) |

| 58 | 1293788 | batch_size = 48
depth (number of layer blocks) = 3
loss_function = MSE + InfoNCE
bilstm_units = 96 | Increase batch size |
|---|---|---|---|
| 59 | 1293788 | batch_size = 64
depth (number of layer blocks) = 3
loss_function = MSE + InfoNCE
bilstm_units = 96 | Increase batch size |
| 60 | 1293788 | batch_size = 32
depth (number of layer blocks) = 3
loss_function = MSE + InfoNCE
bilstm_units = 96 | Dropout rate = 0.2 |
| 61 | 1293788 | batch_size = 64
depth (number of layer blocks) = 3
loss_function = MSE + InfoNCE
bilstm_units = 96 | Increase batch size
Dropout rate = 0.2 |
| 62 | 1293788 | batch_size = 16
depth (number of layer blocks) = 3
loss_function = MSE + InfoNCE
bilstm_units = 96 | Decrease batch size |
| 63 | 1293788 | batch_size = 8
depth (number of layer blocks) = 3
loss_function = MSE + InfoNCE
bilstm_units = 96 | Decrease batch size |
| 64 | 1293788 | batch_size = 80
depth (number of layer blocks) = 3
loss_function = MSE + InfoNCE
bilstm_units = 96 | Increase batch size |
| 65 | 1293788 | batch_size = 32
depth (number of layer blocks) = 3
loss_function = MSE + InfoNCE
bilstm_units = 96 | Dropout rate = 0.15 |
| 66 | 1293788 | batch_size = 32
depth (number of layer blocks) = 3
loss_function = MSE + InfoNCE
bilstm_units = 96 | Dropout rate = 0.05 |
| 67 | 1293788 | batch_size = 32
depth (number of layer blocks) = 3
loss_function = MSE + InfoNCE
bilstm_units = 96 | Dropout rate = 0.25 |
| 68 | 1293788 | batch_size = 32
depth (number of layer blocks) = 3
loss_function = MSE + InfoNCE
bilstm_units = 96 | Dropout rate = 0.30 |
| 69 | 1293788 | batch_size = 32
depth (number of layer blocks) = 3
loss_function = MSE + InfoNCE
bilstm_units = 96 | Dropout rate = 0.35 |

| 70 | 1293788 | batch_size = 48
depth (number of layer blocks) = 3
loss_function = MSE + InfoNCE
bilstm_units = 96 | Increase batch size
Dropout rate = 0.05 |
| 71 | 1293788 | batch_size = 48
depth (number of layer blocks) = 3
loss_function = MSE + InfoNCE
bilstm_units = 96 | Increase batch size
Dropout rate = 0.05
Learning rate =
0.0005 |
| 72 | 1293788 | batch_size = 320
depth (number of layer blocks) = 3
loss_function = MSE + InfoNCE
bilstm_units = 96 | Increase batch size |
| 73 | 1293788 | batch_size = 320
depth (number of layer blocks) = 3
loss_function = MSE + InfoNCE
bilstm_units = 96 | Increase batch size
Dropout rate = 0.05 |
| 74 | 1293788 | batch_size = 480
depth (number of layer blocks) = 3
loss_function = MSE + InfoNCE
bilstm_units = 96 | Increase batch size |
| 75 | 1293788 | batch_size = 320
depth (number of layer blocks) = 3
loss_function = MSE + InfoNCE
bilstm_units = 96 | Increase batch size
Dropout rate = 0.15 |
| 76 | 1293788 | batch_size = 320
depth (number of layer blocks) = 3
loss_function = MSE + InfoNCE
bilstm_units = 96 | Increase batch size
Dropout rate = 0.20 |
| 77 | 1293788 | batch_size = 320
depth (number of layer blocks) = 3
loss_function = MSE + InfoNCE
bilstm_units = 96 | Increase batch size
Dropout rate = 0.25 |
| 78 | 1293788 | batch_size = 320
depth (number of layer blocks) = 3
loss_function = MSE + InfoNCE
bilstm_units = 96 | Increase batch size
Dropout rate = 0.30 |
| 79 | 1293788 | batch_size = 320
depth (number of layer blocks) = 6
loss_function = MSE + InfoNCE
bilstm_units = 96 | Increase depth |
| 80 | 1293788 | batch_size = 32
depth (number of layer blocks) = 3
loss_function = MSE + InfoNCE
bilstm_units = 96 | Exponential decrese
of learning rate
(decay rate = 0.9) |

## C  APPENDIX C

Appendix C provides additional experiments' results.

Table 6: CAE model benchmarking against existing peptide encoding strategies.

| Encoding type | MCC (5-fold cross validation) | | | | | | |
|---|---|---|---|---|---|---|---|
| | ADP | AIP | AMP | AOP | Avg. | Min | Max |
| One-hot | 0.197 (0.015) | 0.216 (0.015) | 0.560 (0.004) | **0.764 (0.013)** | **0.434** | **0.197** | **0.764** |
| Blosum | 0.026 (0.020) | 0.290 (0.012) | 0.337 (0.015) | 0.189 (0.012) | 0.211 | 0.026 | 0.337 |
| Threemers | 0.131 (0.060) | **0.357 (0.009)** | 0.519 (0.003) | 0.539 (0.015) | 0.387 | 0.131 | 0.539 |
| ProtBert | **0.334 (0.031)** | 0.138 (0.021) | **0.658 (0.007)** | 0.580 (0.011) | 0.428 | 0.138 | 0.658 |
| CAE | 0.137 (0.040) | 0.336 (0.023) | 0.421 (0.014) | 0.442 (0.016) | 0.334 | 0.137 | 0.442 |
| MCC performance per task | | | | | | | |
| Avg. | 0.165 | 0.267 | 0.499 | **0.503** | - | - | - |
| Min | 0.026 | 0.138 | **0.337** | 0.189 | - | - | - |
| Max | 0.334 | 0.357 | 0.658 | **0.764** | - | - | - |

Table 7: AE-based architectures benchmarking. Results are presented for best-in-class models.

| Model class | MCC (5-fold cross validation) | | | | | | |
|---|---|---|---|---|---|---|---|
| | ADP | AIP | AMP | AOP | Avg. | Min | Max |
| CAE | **0.137 (0.040)** | 0.336 (0.023) | 0.421 (0.014) | 0.442 (0.016) | **0.334** | **0.137** | 0.442 |
| InfoCVAE | 0.035 (0.033) | 0.019 (0.009) | 0.025 (0.008) | -0.024 (0.028) | 0.014 | -0.024 | 0.035 |
| Transformer-AE | 0.059 (0.024) | 0.122 (0.009) | **0.465 (0.012)** | 0.448 (0.019) | 0.273 | 0.059 | 0.465 |
| BiLSTM-AE | -0.004 (0.053) | **0.342 (0.009)** | 0.415 (0.005) | **0.496 (0.031)** | 0.312 | -0.004 | **0.496** |

Table 8: Classification metrics for four benchmark tasks.

| Peptide property | Encoding type | Accuracy | Precision | Recall | F1 score | ROC AUC | MCC |
|---|---|---|---|---|---|---|---|
| ADP | One-hot | 0.598 (0.008) | 0.596 (0.012) | 0.622 (0.033) | 0.606 (0.009) | 0.597 (0.008) | 0.197 (0.015) |
| | Threemers | 0.559 (0.030) | 0.559 (0.031) | 0.559 (0.041) | 0.558 (0.034) | 0.559 (0.030) | 0.119 (0.060) |
| | Blosum | 0.513 (0.010) | 0.511 (0.008) | 0.560 (0.032) | 0.533 (0.018) | 0.513 (0.010) | 0.026 (0.020) |
| | ProtBERT | 0.665 (0.016) | 0.663 (0.020) | 0.682 (0.038) | 0.669 (0.019) | 0.665 (0.016) | 0.334 (0.031) |
| | cBiLSTM-AE | 0.638 (0.032) | 0.638 (0.034) | 0.652 (0.039) | 0.643 (0.031) | 0.638 (0.032) | 0.277 (0.065) |
| AIP | One-hot | 0.656 (0.005) | 0.608 (0.013) | 0.284 (0.015) | 0.386 (0.016) | 0.585 (0.007) | 0.216 (0.015) |
| | Threemers | 0.704 (0.004) | 0.637 (0.006) | 0.531 (0.014) | 0.579 (0.009) | 0.671 (0.005) | 0.358 (0.009) |
| | Blosum | 0.680 (0.005) | 0.622 (0.013) | 0.420 (0.008) | 0.501 (0.007) | 0.631 (0.005) | 0.290 (0.012) |
| | ProtBERT | 0.627 (0.008) | 0.527 (0.022) | 0.245 (0.012) | 0.335 (0.015) | 0.554 (0.008) | 0.138 (0.021) |
| | cBiLSTM-AE | 0.689 (0.004) | 0.625 (0.008) | 0.468 (0.010) | 0.535 (0.007) | 0.647 (0.004) | 0.316 (0.009) |
| AMP | One-hot | 0.779 (0.002) | 0.810 (0.003) | 0.728 (0.004) | 0.767 (0.002) | 0.779 (0.002) | 0.560 (0.004) |
| | Threemers | 0.759 (0.001) | 0.777 (0.003) | 0.726 (0.002) | 0.751 (0.001) | 0.759 (0.001) | 0.519 (0.003) |
| | Blosum | 0.667 (0.007) | 0.693 (0.010) | 0.601 (0.007) | 0.643 (0.007) | 0.667 (0.007) | 0.337 (0.015) |
| | ProtBERT | 0.828 (0.003) | 0.863 (0.005) | 0.779 (0.004) | 0.819 (0.003) | 0.828 (0.003) | 0.658 (0.007) |
| | cBiLSTM-AE | 0.783 (0.001) | 0.807 (0.006) | 0.746 (0.007) | 0.775 (0.002) | 0.783 (0.001) | 0.569 (0.002) |
| AOP | One-hot | 0.881 (0.007) | 0.889 (0.019) | 0.875 (0.011) | 0.881 (0.005) | 0.881 (0.007) | 0.764 (0.013) |
| | Threemers | 0.769 (0.008) | 0.789 (0.008) | 0.737 (0.016) | 0.761 (0.009) | 0.769 (0.008) | 0.540 (0.015) |
| | Blosum | 0.592 (0.006) | 0.615 (0.006) | 0.494 (0.020) | 0.547 (0.014) | 0.592 (0.006) | 0.189 (0.012) |
| | ProtBERT | 0.790 (0.006) | 0.789 (0.010) | 0.792 (0.010) | 0.790 (0.005) | 0.790 (0.006) | 0.580 (0.011) |
| | cBiLSTM-AE | 0.846 (0.006) | 0.844 (0.006) | 0.848 (0.006) | 0.846 (0.006) | 0.846 (0.006) | 0.692 (0.012) |

Table 9: Regression metrics for four benchmark tasks.

| Peptide property | Encoding type | MAE | RMSE | $R^2$ |
|---|---|---|---|---|
| ISI | One-hot | 15.778 (0.059) | 21.263 (0.177) | 0.445 (0.004) |
| | Threemers | 15.811 (0.049) | 21.279 (0.122) | 0.444 (0.006) |
| | Blosum | 19.637 (0.036) | 26.458 (0.143) | 0.141 (0.010) |
| | ProtBERT | 16.406 (0.059) | 22.578 (0.099) | 0.374 (0.007) |
| | cBiLSTM-AE | 18.021 (0.043) | 24.472 (0.113) | 0.265 (0.011) |
| TNC | One-hot | 1.157 (0.007) | 1.872 (0.019) | 0.823 (0.004) |
| | Threemers | 1.596 (0.009) | 2.443 (0.006) | 0.698 (0.006) |
| | Blosum | 2.830 (0.006) | 4.173 (0.011) | 0.119 (0.011) |
| | ProtBERT | 1.132 (0.005) | 1.731 (0.032) | 0.849 (0.004) |
| | cBiLSTM-AE | 1.200 (0.007) | 1.716 (0.011) | 0.851 (0.003) |
| IEP | One-hot | 1.268 (0.002) | 1.557 (0.004) | 0.694 (0.002) |
| | Threemers | 1.397 (0.007) | 1.745 (0.009) | 0.616 (0.004) |
| | Blosum | 2.272 (0.005) | 2.660 (0.003) | 0.107 (0.003) |
| | ProtBERT | 0.930 (0.003) | 1.231 (0.003) | 0.809 (0.001) |
| | cBiLSTM-AE | 1.069 (0.005) | 1.375 (0.004) | 0.761 (0.002) |
| MW | One-hot | 495.466 (1.437) | 738.626 (3.331) | 0.937 (0.001) |
| | Threemers | 118.957 (1.394) | 180.648 (3.629) | 0.996 (0.0002) |
| | Blosum | 248.439 (1.389) | 363.888 (2.251) | 0.985 (0.0002) |
| | ProtBERT | 552.071 (1.058) | 763.360 (1.980) | 0.932 (0.0004) |
| | cBiLSTM-AE | 75.067 (0.547) | 104.754 (0.715) | 0.999 (0.00002) |

# D APPENDIX D

## D.1 LIMITATIONS

**Sequence length constraints.** During training we capped sequences at 96 residues (more than enough for peptides but small for full-scale proteins) to satisfy the CAE fixed input size. Although subsequent BiLSTM and transformer variants can process variable lengths, the cap was retained for experimental consistency, where markedly longer inputs can render the embedding quality; sequences beyond 96 residues therefore require additional architectural changes.

**Handling of peptide modifications.** The model supports only 20 canonical residues plus a small set of predefined modifications. Extending to user-defined or rare modifications would necessitate (i) quantum-chemical calculations for new DFT descriptors and (ii) updates to the RDKit monomer dictionary.

**Peptide structure limitations.** The framework is restricted to linear peptides; branched or cyclic topologies were excluded. Supporting such structures would entail substantial changes to both preprocessing and model design and is left for future work.

# E    APPENDIX E

## LLM USAGE STATEMENT

Large language models were used during the preparation of this manuscript solely for text refinement and stylistic improvement. Specifically, LLMs assisted with phrasing adjustments and ensuring adherence to academic writing conventions. All scientific content, ideas, and conclusions remain entirely our own.

# F  APPENDIX F

Table 10: Correlation between embedding dimensions and instability index.

| Peptide property | Encoded input feature | PCC (r) |
| --- | --- | --- |
| | exactmw | -0.034 |
| | amw | 0.071 |
| | lipinskiHBA | 0.131 |
| | lipinskiHBD | 0.124 |
| | NumRotatableBonds | 0.010 |
| | NumHBD | 0.017 |
| | NumHBA | 0.101 |
| | NumHeavyAtoms | 0.040 |
| | NumAtoms | -0.120 |
| | NumHeteroatoms | 0.141 |
| | NumAmideBonds | -0.068 |
| | FractionCSP3 | 0.101 |
| | NumRings | -0.071 |
| | NumAromaticRings | -0.253 |
| | NumAliphaticRings | 0.102 |
| | NumSaturatedRings | 0.137 |
| | NumHeterocycles | 0.138 |
| | NumAromaticHeterocycles | -0.013 |
| | NumSaturatedHeterocycles | 0.202 |
| | NumAliphaticHeterocycles | 0.059 |
| | NumSpiroAtoms | -0.193 |
| | NumBridgeheadAtoms | -0.020 |
| ISI | NumAtomStereoCenters | -0.084 |
| | NumUnspecifiedAtomStereoCenters | 0.022 |
| | labuteASA | -0.005 |
| | tpsa | 0.142 |
| | CrippenClogP | -0.097 |
| | CrippenMR | 0.047 |
| | chi0v | -0.057 |
| | chi1v | -0.049 |
| | chi2v | 0.059 |
| | chi3v | 0.069 |
| | chi4v | 0.076 |
| | chi0n | 0.007 |
| | chi1n | -0.036 |
| | chi2n | 0.007 |
| | chi3n | 0.001 |
| | chi4n | 0.016 |
| | hallKierAlpha | -0.017 |
| | kappa1 | 0.045 |
| | kappa2 | 0.063 |
| | kappa3 | 0.056 |
| | Phi | -0.008 |
| | $Ca^{2+}$ interaction energy | 0.068 |
| | $Mg^{2+}$ interaction energy | 0.076 |
| | $Ba^{2+}$ interaction energy | 0.084 |

Table 11: Correlation between embedding dimensions and theoretical net charge.

| Peptide property | Chemical descriptor | PCC (r) |
|---|---|---|
| | exactmw | 0.054 |
| | amw | -0.063 |
| | lipinskiHBA | -0.293 |
| | lipinskiHBD | 0.446 |
| | NumRotatableBonds | 0.065 |
| | NumHBD | 0.012 |
| | NumHBA | 0.006 |
| | NumHeavyAtoms | -0.469 |
| | NumAtoms | 0.206 |
| | NumHeteroatoms | -0.212 |
| | NumAmideBonds | -0.150 |
| | FractionCSP3 | 0.109 |
| | NumRings | -0.332 |
| | NumAromaticRings | -0.320 |
| | NumAliphaticRings | -0.218 |
| | NumSaturatedRings | -0.267 |
| | NumHeterocycles | -0.301 |
| | NumAromaticHeterocycles | -0.055 |
| | NumSaturatedHeterocycles | -0.131 |
| | NumAliphaticHeterocycles | -0.156 |
| | NumSpiroAtoms | -0.238 |
| | NumBridgeheadAtoms | -0.006 |
| | NumAtomStereoCenters | -0.537 |
| TNC | NumUnspecifiedAtomStereoCenters | -0.033 |
| | labuteASA | 0.074 |
| | tpsa | 0.012 |
| | CrippenClogP | -0.738 |
| | CrippenMR | 0.075 |
| | chi0v | 0.179 |
| | chi1v | -0.055 |
| | chi2v | 0.013 |
| | chi3v | 0.032 |
| | chi4v | 0.022 |
| | chi0n | 0.254 |
| | chi1n | 0.147 |
| | chi2n | -0.140 |
| | chi3n | -0.109 |
| | chi4n | -0.140 |
| | hallKierAlpha | 0.190 |
| | kappa1 | 0.035 |
| | kappa2 | 0.395 |
| | kappa3 | 0.042 |
| | Phi | 0.402 |
| | $Ca^{2+}$ interaction energy | 0.542 |
| | $Mg^{2+}$ interaction energy | 0.369 |
| | $Ba^{2+}$ interaction energy | 0.375 |

Table 12: Correlation between embedding dimensions and isoelectric point.

| Peptide property | Chemical descriptor | PCC (r) |
|---|---|---|
| | exactmw | 0.012 |
| | amw | -0.005 |
| | lipinskiHBA | -0.150 |
| | lipinskiHBD | 0.387 |
| | NumRotatableBonds | 0.028 |
| | NumHBD | -0.060 |
| | NumHBA | 0.022 |
| | NumHeavyAtoms | -0.334 |
| | NumAtoms | 0.197 |
| | NumHeteroatoms | -0.091 |
| | NumAmideBonds | -0.082 |
| | FractionCSP3 | 0.147 |
| | NumRings | -0.335 |
| | NumAromaticRings | -0.321 |
| | NumAliphaticRings | -0.121 |
| | NumSaturatedRings | -0.216 |
| | NumHeterocycles | -0.185 |
| | NumAromaticHeterocycles | 0.030 |
| | NumSaturatedHeterocycles | -0.064 |
| | NumAliphaticHeterocycles | -0.162 |
| | NumSpiroAtoms | -0.215 |
| | NumBridgeheadAtoms | 0.061 |
| IEP | NumAtomStereoCenters | -0.392 |
| | NumUnspecifiedAtomStereoCenters | 0.043 |
| | labuteASA | 0.015 |
| | tpsa | 0.079 |
| | CrippenClogP | -0.620 |
| | CrippenMR | 0.029 |
| | chi0v | 0.110 |
| | chi1v | -0.112 |
| | chi2v | 0.076 |
| | chi3v | 0.062 |
| | chi4v | 0.033 |
| | chi0n | 0.220 |
| | chi1n | 0.084 |
| | chi2n | -0.130 |
| | chi3n | -0.069 |
| | chi4n | -0.135 |
| | hallKierAlpha | 0.150 |
| | kappa1 | 0.095 |
| | kappa2 | 0.359 |
| | kappa3 | 0.105 |
| | Phi | 0.367 |
| | $Ca^{2+}$ interaction energy | 0.494 |
| | $Mg^{2+}$ interaction energy | 0.365 |
| | $Ba^{2+}$ interaction energy | 0.371 |

Table 13: Correlation between embedding dimensions and molecular weight.

| Peptide property | Chemical descriptor | PCC (r) |
| --- | --- | --- |
| | exactmw | -0.601 |
| | amw | 0.825 |
| | lipinskiHBA | 0.725 |
| | lipinskiHBD | 0.192 |
| | NumRotatableBonds | -0.281 |
| | NumHBD | -0.948 |
| | NumHBA | 0.726 |
| | NumHeavyAtoms | 0.730 |
| | NumAtoms | -0.090 |
| | NumHeteroatoms | 0.828 |
| | NumAmideBonds | 0.472 |
| | FractionCSP3 | 0.154 |
| | NumRings | -0.294 |
| | NumAromaticRings | -0.196 |
| | NumAliphaticRings | 0.546 |
| | NumSaturatedRings | -0.368 |
| | NumHeterocycles | 0.327 |
| | NumAromaticHeterocycles | 0.927 |
| | NumSaturatedHeterocycles | 0.176 |
| | NumAliphaticHeterocycles | -0.740 |
| | NumSpiroAtoms | -0.556 |
| | NumBridgeheadAtoms | 0.819 |
| | NumAtomStereoCenters | 0.138 |
| MW | NumUnspecifiedAtomStereoCenters | 0.979 |
| | labuteASA | -0.451 |
| | tpsa | 0.774 |
| | CrippenClogP | -0.257 |
| | CrippenMR | -0.272 |
| | chi0v | -0.687 |
| | chi1v | -0.994 |
| | chi2v | 0.992 |
| | chi3v | 0.766 |
| | chi4v | 0.386 |
| | chi0n | -0.350 |
| | chi1n | -0.488 |
| | chi2n | -0.040 |
| | chi3n | 0.434 |
| | chi4n | -0.006 |
| | hallKierAlpha | -0.257 |
| | kappa1 | 0.849 |
| | kappa2 | 0.685 |
| | kappa3 | 0.989 |
| | Phi | 0.732 |
| | $Ca^{2+}$ interaction energy | 0.734 |
| | $Mg^{2+}$ interaction energy | 0.888 |
| | $Ba^{2+}$ interaction energy | 0.883 |

Table 14: Regression tasks results.

| Encoding type | $R^2$ (5-fold cross validation) | | | | | | |
|---|---|---|---|---|---|---|---|
| | ISI | TNC | IEP | MW | Avg. | Min | Max |
| One-hot | **0.445** **(0.004)** | 0.823 (0.004) | 0.694 (0.002) | 0.937 (0.001) | 0.725 | **0.445** | 0.937 |
| Blosum | 0.141 (0.010) | 0.119 (0.011) | 0.107 (0.003) | 0.985 (0.000) | 0.338 | 0.107 | 0.985 |
| Threemers | 0.444 (0.006) | 0.698 (0.006) | 0.616 (0.004) | 0.996 (0.000) | 0.689 | 0.444 | 0.996 |
| cBiLSTM-AE | 0.265 (0.011) | **0.851** **(0.003)** | 0.761 (0.002) | **0.999** **(0.000)** | 0.719 | 0.265 | **0.999** |

