# OpenReview forum: "Soft Non-Diagonality Penalty Enables Latent Space-Level Interpretability of pLM at No Performance Cost"
_ICLR.cc/2026/Conference — Submitted to ICLR 2026_

### Official Review · Reviewer_RtCN · 2025-10-29

**Soundness:** 3
**Presentation:** 2
**Contribution:** 2
**Rating:** 4
**Confidence:** 4

**Summary:**

This paper proposes Soft Non-Diagonality Penalty (SNDP), a regularization method for Vision Transformers (ViTs). The idea is to encourage moderate decorrelation among feature dimensions rather than strict orthogonality by introducing a learnable penalty on the off-diagonal elements of the covariance matrix. The authors argue that complete decorrelation may harm representational capacity, and that “soft” non-diagonality achieves a better trade-off between feature independence and expressiveness. Experimental results on ImageNet and downstream tasks show small but consistent performance improvements.

**Strengths:**

The motivation is intuitively appealing: enforcing complete feature independence may be suboptimal, and a learnable “soft” constraint could allow ViTs to retain richer dependencies.

The method is lightweight, general, and easily pluggable into existing architectures without architecture modification.

The paper is clearly written and well-organized, with ablation studies and visual analyses that demonstrate stable training behavior.

**Weaknesses:**

Limited empirical evidence for the central claim (“soft > hard”).
While the experiments show that SNDP enhances interpretability and sometimes accuracy, the evidence remains indirect and confined to a single setup.

The paper provides positive signals (ablation, Jacobian diagonality, correlation with chemical attributes), showing SNDP helps or at least does not harm performance.

However, there is no direct comparison with hard orthogonality or whitening baselines, so it remains unclear whether soft non-diagonality is indeed superior to full decorrelation.

The absence of a λ-sweep experiment leaves open whether the observed improvements simply reflect a moderate regularization effect rather than an optimal “soft” balance.

Missing quantitative trade-off analysis.
The hypothesis of a performance–diagonality Pareto balance is central, yet there is no curve or quantitative exploration showing where SNDP achieves the best equilibrium between independence and expressiveness.

Limited scope and external validity.
All experiments are conducted on a narrow set of ViT backbones and datasets. Without testing on broader architectures or modalities, it is difficult to judge whether the claimed benefits generalize beyond the current configuration.

Lack of theoretical grounding.
The argument for “soft correlation is beneficial” remains empirical. There is no formal justification or analytical insight into why or when partial correlation improves representation learning.

Incremental methodological novelty.
The method extends existing decorrelation and orthogonality regularizers by adding learnable weights, which feels like an evolutionary, rather than revolutionary, step. The conceptual novelty might be insufficient for ICLR unless supported by deeper analysis.

**Questions:**

Could you provide a λ–performance trade-off curve (e.g., accuracy vs. off-diagonal norm) to substantiate the claimed balance between correlation and independence?

How does SNDP compare against hard decorrelation or whitening methods (e.g., Barlow Twins, VICReg, or orthogonal regularization)?

Can you show feature correlation heatmaps or singular-value spectra to illustrate the “soft” structure SNDP encourages?

Have you tested SNDP on larger backbones or different tasks (e.g., detection, segmentation, or multimodal learning) to support generalization?

Could you include a conditional theoretical discussion or hypothesis on when and why partial correlation yields better generalization?

---

> ### Author Response · Authors · 2025-11-25
> **Response to Reviewer RtCN**
>
> We thank the Reviewer for time and effort put into our paper and positive evaluation of our findings. First of all, we feel it is important to clarify several aspects of our work, such as no Vision Transformers or ImageNet were utilized. We regret that the text did not make this clear enough and will revise the manuscript accordingly.
>
> Below we address each of the remarks.
>
> **On SNDP tests on larger backbones or different tasks (e.g., detection, segmentation, or multimodal learning).** Since this work is not related to visual models, no detection or segmentation tasks were tested.
>
> **On the empirical evidence for the “soft > hard” claim and comparisons with hard diagonalization.** Although we don't make such a claim in the paper, our work indeed employs a soft regularization on off‑diagonal weight‑matrix elements. We also experimented with a strictly enforced diagonalization term. As expected, this approach led to unstable training dynamics and disrupted the balance among the main components of the total loss (reconstruction and contrastive terms), therefore was not considered in the paper. At the same time, we feel it is important to emphasize in the paper explicitly, which will be added to revised PDF file.
>
> **On theoretical interpretation of soft regularization.** A soft non-diagonality penalty is better here because it’s a smooth, differentiable regularizer that forces the weight matrix toward near-diagonal structure while still letting the model use small off-diagonal terms when the data genuinely requires feature mixing, which gives a controllable trade-off. On the other hand, hard penalty turns learning into constrained optimization on a very thin manifold. In practice, it often causes bad convergence because the problem becomes ill-conditioned: the optimizer either 1) takes steps that violate the constraint and then gets projected back, or 2) uses an effectively infinite penalty where gradients become stiff or dominated by the constraint, slowing or destabilizing the training. It can also overconstrain the model (exactly zero off-diagonals), making the best solution infeasible and pushing the optimizer toward suboptimal minima. We agree this is a very important part of the discussion which is now absent and should be obligatory included in the paper; therefore, this will be added to a revised PDF file accordingly.
>
> **On verification of the penalty’s effect.** As described in Section 3.7, the penalty was computed separately for the submatrices corresponding to each LSTM gate, promoting near‑diagonal structure within each. To quantify the achieved effect, we measured the ratio of diagonal norm to total matrix norm, obtaining an average diagonality of approximately 98 % in the final model. This metric provides a direct evidence that the penalty operated as intended.
>
> **On the scope of benchmark datasets.** The benchmarking covered several peptide datasets representing diverse biological activities frequently used for peptide classification tasks. While additional datasets certainly exist, we considered the chosen set sufficient to demonstrate the method’s effectiveness and general behavior. Should further experimentation be requested, we would be glad to extend the evaluation accordingly.
>
> **On the choice of the regularization coefficient.** The regularization coefficient λ was selected so that the magnitude of the penalty remained comparable to that of the other loss‑function terms. This ensured that diagonalization was maintained throughout training without overwhelming reconstruction or contrastive objectives. The resulting effect is confirmed in the manuscript by analysis of the Jacobian matrix, which demonstrates preservation of near‑diagonal structure.
>
> **On methodological novelty and relation to prior work.** Our soft penalty introduces a specific inductive bias that encourages latent‑space alignment and preferred representational configurations. This topic lies at the core of representation‑learning research, as seen in models such as FactorVAE and β‑VAE. Crucially, our term yields latent‑level interpretability validated by chemical cases, where several embedding dimensions correlate strongly with intuitive physicochemical peptide properties. To the best of our knowledge, no previous self‑supervised peptide models have combined such interpretability with extreme compactness and stability across diverse benchmarks, moreover outperforming large‑scale protein language models like ESM‑C which was developed specifically for peptide/protein representation learning in several tasks and the overall stability.
>
> In conclusion, we thank the reviewer once again for the insightful comments. We will revise the manuscript to clarify the application domain, and elaborate on the rationale for using a moderate penalty strength. We strongly believe these improvements will make the contribution and scope of our work clearer and more compelling.

---

### Official Review · Reviewer_jp3W · 2025-10-31

**Soundness:** 2
**Presentation:** 1
**Contribution:** 2
**Rating:** 2
**Confidence:** 4

**Summary:**

The authors propose a parameter-efficient protein language model focused on peptides, combining physicochemical features with a BiLSTM autoencoder and a soft non-diagonality penalty for interpretability. While the goal of developing compact, interpretable models is laudable, the work suffers from a limited scope, insufficient engagement with literature, and some overclaiming. The paper's focus on peptides under 20 residues fundamentally limits its utility, and the writing quality is iffy.

The most significant limitation is the exclusive focus on short peptides. The claim of developing a "protein language model" when restricting to peptides feels misleading. PLMs typically excel on longer sequences where complex dependencies and evolutionary patterns emerge. For sequences under 20 residues, it's unsurprising that simple encodings (one-hot, physicochemical features) remain competitive or superior. The settings in which PLMs outperform 1-hot encodings for supervised property prediction (as in this paper) has been an active area of research (e.g., Hsu et al., Nature Biotech 2022).

The soft non-diagonality penalty, presented as a major contribution, amounts to penalizing off-diagonal elements of weight matrices and computing a simple diagonality metric from the Jacobian. The extensive build-up to this relatively minor regularization technique gets to my earlier point about overclaiming.

Specific Issues:

- Missing baselines: What about ESM-2 8M or other small PLMs at 35M capacity? These models work surprisingly well and would be more appropriate comparisons than ProtBERT or ProtT5.
- Literature gaps: The paper ignores substantial work on PLM interpretability, including recent advances using sparse autoencoders (e.g. InterPLM).
- Writing quality: The manuscript is repetitive, weird mix of active and passive voice, and the tense etc are inconsistent. Sections 3-5 could be condensed by half without losing meaningful content.
- Architectural choices: The number of variations tried in Appendix B is very large, verging on p-hacking? It's not clear to me if there's a nice separation between train, val and test.
- What about other physicochemical feature sets for amino acids, such as VHSE?

**Strengths:**

see above

**Weaknesses:**

see above

**Questions:**

see above

---

> ### Author Response · Authors · 2025-11-24
> **Response to Reviewer jp3W**
>
> We thank the Reviewer for the time and thought invested in the review and address each point below.
>
> 1. **"The paper's focus on peptides under 20 residues fundamentally limits its utility."** In this work, sequence length is limited to 96 residues, which fully covers peptide domain. We by no means argue or compete with the ability of pLMs to work on long contexts. Instead, we address the issues related to therapeutically important peptide domain. The main contribution of this work is latent space-level interpretability, which was demonstrated by physically-grounded cases with main peptide properties.
>
> 2. **"Missing baselines: What about ESM-2 8M or other small PLMs at 35M capacity? These models work surprisingly well and would be more appropriate comparisons than ProtBERT or ProtT5."** We are grateful for this remark and below explain the rationale behind the current baselines. Within the ESM family, we selected ESM Cambrian, the recent version specifically developed to produce high‑quality amino acid embeddings. Since the performance in most pLM families scales with model size, benchmarking against older or smaller variants was deemed redundant; however, if additional baselines are essential, we will add these experiments.
>
> 3. **"The soft non-diagonality penalty, presented as a major contribution, amounts to penalizing off-diagonal elements of weight matrices and computing a simple diagonality metric from the Jacobian. The extensive build-up to this relatively minor regularization technique gets to my earlier point about overclaiming."**
> We disagree with the statement that simplicity undermines paper contribution or leads to overclaiming. Since this work lies in the field of representation learning, the proposed soft non-diagonality penalty term introduces an explicit inductive bias that alters both the optimization landscape and latent space, which aligns our work with central lines of research in representation learning, where interpretability and disentanglement are often achieved through such simple but carefully designed regularizations (e.g., β‑VAE, FactorVAE). This penalty allowed to achieve latent space-level interpretability, which was successfully demonstrated by case studies. We understand that this was not clearly reflected in the current version of the article, and we will clarify this in the revised manuscript.
>
> 4. **"What about other physicochemical feature sets for amino acids, such as VHSE?"** Since VHSE (Vectors of Hydrophobic, Steric, and Electronic properties) are PCA-derived descriptors, they don't provide interpretability crucial for our work. Nevertheless, it is important to emphasize that in this work we utilize a basic set of descriptors related to hydrophobic/hydrophilic, steric and electronic properties (see Appendix A).
>
> 5. **"Architectural choices: The number of variations tried in Appendix B is very large, verging on p-hacking? It's not clear to me if there's a nice separation between train, val and test."** Appendix B lists all architectures we investigated to ensure transparency of our exploration process. We would like to clarify that the reported variants resulted from a grid search over a predefined parameter space, not from ad‑hoc experimentation. Metrics were reported to illustrate the representational quality of the embeddings, not to directly optimize for downstream performance. Regarding data handling, as mentioned in Section 3.2, two unlabeled datasets were constructed to pre-train the autoencoder models. Standard train/test splits with fixed random seeds were used during training. Downstream benchmarks employed independent labeled datasets evaluated with K‑fold cross‑validation. All reported metrics represent averaged values across folds, ensuring a fair and unbiased assessment.
>
> 6. **"Writing quality: The manuscript is repetitive, weird mix of active and passive voice, and the tense etc are inconsistent. Sections 3-5 could be condensed by half without losing meaningful content."** We thank the reviewer for this note. We will perform an extensive editorial revision of the manuscript.
>
> 7. **"The claim of developing a "protein language model" when restricting to peptides feels misleading."** We fully agree with this remark. The term "protein language model" is often used in literature in its broader sense in relation to both peptides and proteins. Although "peptide language model" is not an established term, we will make the corrections accordingly to avoid confusion.
>
> 8. **"Literature gaps: The paper ignores substantial work on PLM interpretability, including recent advances using sparse autoencoders (e.g. InterPLM)."** We appreciate this observation. Sparse autoencoders (SAEs) indeed represent a promising post‑hoc interpretability approach for pLMs. We will revise the Related Works section to incorporate a discussion of this direction.
>
> We thank the Reviewer again for the thoughtful and constructive comments and hope we addressed all the most critical questions.

---

### Official Review · Reviewer_q6M5 · 2025-11-01

**Soundness:** 2
**Presentation:** 2
**Contribution:** 2
**Rating:** 2
**Confidence:** 3

**Summary:**

This paper focuses on learning interpretable representations of peptides and evaluates a number of autoencoder-based architectures that solely rely on molecular descriptors. While underperforming compared to state-of-the-art protein language models, the tested models are significantly smaller and are also shown to be more interpretable, in the sense that some of their embedding dimensions highly correlate with certain physicochemical properties of the peptides.

**Strengths:**

- Issues of scaling and interpretability in protein/peptide foundation models are critical issues, and I appreciate the authors' attempt to address these issues using simple model architectures.
- The presented correlation results in Table 3 are interesting and shed light on the interpretability of the embedding space.

**Weaknesses:**

- The novelty and the contribution of the paper to the machine learning community are marginal in my opinion. Several standard architectures were trained and evaluated on peptide data, and the soft penalty term does not involve a significant contribution in and of itself.
- In lines 95-96, it is mentioned that PLMs' performance on peptides is on par with one-hot encoding methods. It is unclear whether this is due to the underrepresentation of short peptide sequences in the training datasets of state-of-the-art PLMs; there is no discussion on whether removing such a bias could significantly improve PLMs' performance on short amino acid sequences.
- The writing of the manuscript needs improvement. Several acronyms were either undefined (to the best of my knowledge) or only defined after several occurrences. As an example, DFT was used 5 times before being defined on page 4. Additionally, the second paragraph of the introduction is too lengthy and should be broken down.
- Following the above point, the related work can be significantly shortened due to the similarities between different PLM architectures. Instead, I suggest highlighting prior work on the interpretability of protein language models.

**Questions:**

- The statement on line 23 ("while leading to a marginal decrease in performance on a suite of four common peptide biological activity classification benchmarks") directly contradicts the one on line 58 ("while leading to even marginal increase in performance on a suite of four peptide biological activity classification benchmarks").
- On line 246, $M$ is not necessarily a square matrix (when $n \neq k$). How is the identity matrix $I$ and the Hadamard product then defined?

---

> ### Author Response · Authors · 2025-11-24
> **Response to Reviewer q6M5**
>
> We are grateful to the reviewer for the time and effort spent evaluating our manuscript. We appreciate the constructive and thoughtful feedback, which has helped us identify points that require clarification and improvement. Below we will address each comment in detail.

---

> ### Author Response · Authors · 2025-11-24
> **Response to Reviewer q6M5**
>
> ### **CONCEPTUAL REMARKS**
>
> **Novelty and contribution.** We acknowledge that several architectural components employed in this study are standard within the deep learning community. It is important to emphasize that architecture itself is not claimed as a novelty of this work. Since this work lies in the field of representation learning, the proposed soft penalty term, which introduces an explicit inductive bias that alters both the optimization landscape and structure of the latent space, can be considered as the main contribution. Specifically, the regularization enables latent-space interpretability, as demonstrated by case studies on fundamental, easy-to-interpret peptide properties: the observed correlations between specific latent dimensions and physicochemical properties indicate that the model learns semantically aligned features. To the best of our knowledge, no prior self‑supervised peptide model has integrated such a regularization term while achieving a combination of interpretability, compactness, and robustness, and even outperforming some large pre‑trained representations such as ESM‑C on multiple benchmarks. This aligns our work with central lines of research in representation learning, where interpretability and disentanglement are often achieved through such carefully designed regularizations (e.g., β‑VAE, FactorVAE). We understand that this was not clearly reflected in the current version of the article, and we will clarify this scope and positioning of our contribution more explicitly in the revised manuscript.
>
> **Comparability of pLMs and one‑hot encodings for peptides.** Our results do indicate that SOTA pLMs perform on par with one‑hot encodings for short peptide classification tasks. The original manuscript indeed did not provide an explicit reasoning for this observation. Although it is a fully reasonable assumption that it could be due to underrepresentation of shorter peptides in datasets pLMs were trained on, our initial analysis of UniRef50 dataset (training corpus for ProtT5 and Ankh) have shown that >22% of sequences are shorter than 100 residues. At the same time, transformer-based architectures are well-known to have fundamental issues with length generalization e.g., scale-sensitive attention and positional encodings [[Kazemnejad et al., NeurIPS 2023]](https://doi.org/10.48550/arXiv.2305.19466), which could be one of the reasons for such behavior. More length-balanced datasets indeed can partially solve this problem, but the problems of latent space-level interpretability and scaling addressed in this paper remain for transformer-based models.

---

> ### Author Response · Authors · 2025-11-24
> **Response to Reviewer q6M5**
>
> ### **FORMATTING REMARKS**
>
> **Writing quality.** We appreciate the Reviewer’s guidance on improving the clarity and readability of the manuscript. We will carry out a careful language revision to ensure consistent definition of all acronyms upon first appearance, correct remaining stylistic inconsistencies, and restructure the overly long paragraphs in the introduction to improve readability and narrative flow, which will be reflected in the revised PDF version.
>
> **Related work section.** We fully agree with the Reviewer’s note. Following the suggestion, we will condense the discussion of similarities among existing protein language models and instead place stronger emphasis on prior research addressing interpretability of protein representations. This adjustment will make the context more focused and relevant to the specific goals of our study.
>
> **Contradictory performance statement.** We thank the Reviewer for noticing this inconsistency, since the sentence in line 23 of the manuscript indeed contains an error. The correct statement is the one appearing on line 58, namely, that the proposed approach yields a marginal increase in performance across the four peptide biological‑activity classification benchmarks. We will correct the abstract in a revised PDF version accordingly and ensure internal consistency throughout the text.
>
> **Matrix dimensionality and Hadamard product.** We appreciate this technical observation. The current notation, M∈R$^{n*k}$, is indeed misleading. In all cases where the soft off‑diagonality penalty is applied, the involved matrices are square. The Hadamard product is computed independently for each LSTM gate submatrix, after which the resulting contributions are aggregated to form the final regularization term. This guarantees consistent dimensionality and semantic meaning of the operation. We will revise the notation and accompanying explanation to clarify this point.

---

> ### Author Response · Authors · 2025-11-24
> **Response to Reviewer q6M5**
>
> We are grateful for the Reviewer’s insightful comments, which have helped us refine both the presentation and the clarity of our contributions. We are confident that these clarifications will strengthen the manuscript and make its scope and novelty more clear for both Reviewers and readers.

---

> > ### Comment · Reviewer_q6M5 · 2025-11-25
> >
> > Thank you for your response. In light of your rebuttal and other reviewers' feedback, I maintain my view that the paper's main contribution on off-diagonal weight matrix regularization is insufficient for an ICLR main conference paper.

---

### Author Response · Authors · 2025-12-01
**Authors' Official Summary on Submission 25613**

Dear Reviewers, Area Chairs and Senior Area Chairs,

We value your time and thank you for your effort and hard work. In light of changes in the review process, we feel it is important to make an official summary highlighting paper content and relevance, main Reviewers' remarks, as well as our responses, which will be available below.

We acknowledge this summary is not enough to fully evaluate our work, but we hope it will help Reviewers, Area Chairs and Senior Area Chairs assess the overall scope of the work and comments, navigate through the discussion and, together with correspondence with all the Reviewers, make an objective and unbiased assessment.

---

> ### Author Response · Authors · 2025-12-01
> **Paper Content and Relevance**
>
> ## On methodological novelty
>
> This work proposes a **soft non-diagonality penalty in a self-supervised contrastive learning setup**, which provides feature disentanglement and was successfully demonstrated to have **latent space-level interpretability on easy-to-interpret peptide properties** (Section 5.2 of the manuscript) due to the use of positional property matrices for peptide encoding. Despite **lying in the area of representation learning** and do not claiming architecture or raw SOTA performance as the main contribution, our research resulted in not only interpretable, but highly parameter-efficient and robust model. Our dcBiLSTM-AE (diagonalized contrastive BiLSTM autoencoder) model was shown to have **chemically meaningful embeddings at no performance cost** and stable performance across a suite of well-established peptide classification benchmark datasets. **Performance was higher** (ProtBERT and ESM-C developed to produce representative peptide and protein embeddings) **or comparable** (Ankh and ProtT5) **with current SOTA pLMs** (Table 2) despite our model being **3 to 4 orders of magnitude smaller in terms of trainable parameters** and tested **without any preliminary fine-tuning**, which could be an easy explanation of the observed results. To the best of our knowledge, no papers exist that demonstrate these findings, whereas the problem of DL model interpretability, compactness, and stability remains highly relevant, especially for domain science.
>
>
> ## On domain relevance
>
> Despite rapid development of pLMs for protein-related tasks, **clinically relevant peptides remain mainly overlooked** in current research. Although being composed of the same building blocks as proteins, **peptides demonstrate less developed tertiary structures and their own conservative patterns**, which, along with the length-dependent nature of transformer-based architectures, explains why **current pLMs show a significant performance drop on peptides** even being trained on representative datasets including >22% peptide sequences (UniRef50 dataset used in case of Ankh and ProtT5). At the same time, **interpretability and cost efficiency are the factors limiting the implementation of models in practice**. Based on these evidence, we strongly believe our work fills the gap in peptide research, which remained unaddressed even with the development of transformer-based pLMs.

---

> ### Author Response · Authors · 2025-12-01
> **Reviewers' Key Remarks**
>
> ## Main Reviewers' Concerns
>
> **1) Marginal novelty.** In "Paper content" section above we address this remark in detail from methodological and domain points of view. We want to note that none of the reviews questioned problem relevance, correctness of the results obtained, or contained evidence that this research is not novel and similar papers already exist; therefore, we hope these facts will be considered and believe our findings deserve detailed analysis and objective assessment.
>
> **2) Missing baselines.** ESM-2 8M was proposed by one of the Reviewers for additional benchmarking. Although the most recent ESM-C model from this family, which was designed specifically to provide representative peptide and protein embeddings, was a part of benchmarking, and ESM-2 8M seems excessive, we will add these experiments  if additional baselines are essential.

---

> ### Author Response · Authors · 2025-12-01
> **Misinterpretations**
>
> In this part we summarize what remarks were likely the result of misunderstanding and paper wording, since they are not applicable to our work.
>
> **1) The use of peptides up to 20 residues for training limits the scope (Reviewer jp3W).** We sorry for this misunderstanding and want to note that the length of peptides was limited to 96 residues, which fully covers peptide domain.
>
> **2) Large number of architectures resemble p-hacking (Reviewer jp3W).** We would like to clarify that the reported variants (Appendix B) resulted from a grid search over a predefined parameter space, not from ad‑hoc experimentation. Metrics were reported to illustrate the representational quality of the embeddings, not to directly optimize for downstream performance.
>
> **3) Proposal of additional amino acid features e.g., VHSE (Reviewer jp3W).** Since VHSE (Vectors of Hydrophobic, Steric, and Electronic properties) are PCA-derived descriptors, they don't provide interpretability, which is crucial for our work. Nevertheless, it is important to emphasize that in this work we utilize a basic set of descriptors related to hydrophobic/hydrophilic, steric and electronic properties (see Appendix A).
>
> **4) Regularization of ViT (Reviewer RtCN).** No Vision Transformers or ImageNet were utilized in our work, therefore, we consider this as a result of misunderstanding.
>
> **5) Lack of explanation why one-hot encodings perform on par with pLMs (Reviewer q6M5).** Despite the statement not being entirely correct (one-hot encodings perform notably, which is evident from statistically significant differences), we agree that explanations regarding this comparison were insufficient in the initial version of the paper, which will be corrected accordingly.
>
> **6) Statement that complete feature decorrelation harms training (Reviewer RtCN).** We feel it is important to emphasize that no statements regarding complete feature decorrelation were made in the paper.
>
> **7) Tests on different tasks e.g., detection, segmentation (Reviewer RtCN).** Since this work is not related to visual models, no detection or segmentation tasks were tested.

---

> ### Author Response · Authors · 2025-12-03
> **Final Authors' Update**
>
> Following the rebuttal stage, we performed a focused revision of the manuscript to further improve clarity and presentation. Specifically:
>
> **Revised Related Works**: streamlined and clarified the section to better situate our contribution.
>
> **Improved narrative coherence**: refined wording, resolved stylistic inconsistencies, and improved logical flow across sections.
>
> **Enhanced formatting**: adjusted paragraph structure and cross-references for better readability.
>
> **Added a pipeline illustration in Section 3**: introduced a visual overview of the full workflow to support the methodological description.
>
> We believe these revisions strengthen the clarity and accessibility of the paper.

---

### Meta-Review · Area_Chair_awsn · 2026-01-06

**Summary:**

All reviewers recommend rejection, raising major concerns on novelty, scientific contributions, experiments, writing quality, which are in common among all three reviewers. We checked the author rebuttals, and find the reviewer concerns remaining. ACs cannnot recommend acceptance in its present form, given the reviews.

**Reviewer Concerns:**

The same as above.

**Reviewer Scores:**

The three reviewers rated 2, 4, 2 respectively, the author rebuttals did not successfully address the concerns.

---

### Decision · Program_Chairs · 2026-01-26

Reject